# The Advanced Lipoxidation End-Product Malondialdehyde-Lysine in Aging and Longevity

**DOI:** 10.3390/antiox9111132

**Published:** 2020-11-15

**Authors:** Mariona Jové, Natàlia Mota-Martorell, Irene Pradas, Meritxell Martín-Gari, Victoria Ayala, Reinald Pamplona

**Affiliations:** Department of Experimental Medicine, Lleida Biomedical Research Institute (IRBLleida), Lleida University (UdL), 25198 Lleida, Catalonia, Spain; Mariona.jove@udl.cat (M.J.); nataliamotamartorell@gmail.com (N.M.-M.); irene.pradas@gmail.com (I.P.); meritxell.martin@udl.cat (M.M.-G.); victoria.ayala@udl.cat (V.A.)

**Keywords:** advanced lipoxidation end-products, aging, carbonyl-amine reaction, cytotoxicity, dietary restriction, longevity, metabolism, reactive carbonyl species

## Abstract

The nonenzymatic adduction of malondialdehyde (MDA) to the protein amino groups leads to the formation of malondialdehyde-lysine (MDALys). The degree of unsaturation of biological membranes and the intracellular oxidative conditions are the main factors that modulate MDALys formation. The low concentration of this modification in the different cellular components, found in a wide diversity of tissues and animal species, is indicative of the presence of a complex network of cellular protection mechanisms that avoid its cytotoxic effects. In this review, we will focus on the chemistry of this lipoxidation-derived protein modification, the specificity of MDALys formation in proteins, the methodology used for its detection and quantification, the MDA-lipoxidized proteome, the metabolism of MDA-modified proteins, and the detrimental effects of this protein modification. We also propose that MDALys is an indicator of the rate of aging based on findings which demonstrate that (i) MDALys accumulates in tissues with age, (ii) the lower the concentration of MDALys the greater the longevity of the animal species, and (iii) its concentration is attenuated by anti-aging nutritional and pharmacological interventions.

## 1. Introduction

An enzymatic post-translational modification (PTM) is a chemical modification of one or more amino acids of a protein in a given biological system [1]. These modifications can be, either irreversible or reversible. Examples of PTMs are protein acetylation, glycosylation, methylation, phosphorylation, sumoylation, and ubiquitylation. Indeed, several hundred types of enzymatic PTMs have been described as affecting a significant portion of the cell proteome [2]. PTMs alter the structure and function of proteins in the cells [1]. All cells of living organisms utilize the PTMs to control their signaling networks and physiological processes, further expanding their protein functions. Among these functions are: Determination or regulation of catalytic activity, interaction with ligands, protein-protein interaction, protein folding, protein turnover, signaling function, and targeting specific subcellular compartments [1,3,4].

In contrast, a non-enzymatic PTM is a chemical modification, reversible or irreversible, mediated by reactive compounds on one or more amino acids of proteins, but as an inescapable event of endogenous chemical cell damage. Examples of non-enzymatic PTMs are glycation, glycoxidation, nitrosylation, oxidation, succination, and lipoxidation [5,6,7]. In this review article, we will focus on the chemical adduction of the reactive compound malondialdehyde to lysine residues in proteins, designated as lipoxidation, its meaning in biology, and its putative role as indicator of aging and longevity.

## 2. The Protein Adduct Malondialdehyde-Lysine

The reaction of malondialdehyde (MDA) with the amino group of the side chain of lysine (Lys) residues in proteins via a Schiff base reaction (reversible covalent adduct) leads to the formation of the malondialdehyde-lysine (MDALys) adduct and the lys-MDA-lys cross-link [8,9,10,11,12,13,14,15]. MDA can also generate a Lys fluorescent adduct and Arg–Lys cross-link adduct [10,13,16,17,18]. This non-enzymatic reaction is called protein lipoxidation [5] and the generated products are called Advanced Lipoxidation End-products (ALEs) [5,19]. Figure 1 shows the formation of the MDALys adduct and the lys-MDA-lys cross-link by the reaction of malondialdehyde with lysine residues in proteins.

Although, MDA can be also generated enzymatically as a byproduct of the cyclooxygenase reaction in thromboxane and prostaglandin biosynthesis, this compound results mainly from the oxidative degradation of polyunsaturated fatty acids (PUFAs), being arachidonic (20:4n6) and docosahexaenoic (22:6n3) acids the main precursors [20]. In biological systems, MDA is a product of lipid peroxidation of cell membranes as a consequence of reaction of PUFAs and radical species [20,21,22,23]. Therefore, lipid peroxidation generates hydroperoxides, which undergo fragmentation to produce the reactive intermediate with three carbons in length called MDA [20]. More specifically, MDA is a reactive di-aldehydes (alkanedial) characterized by two carbonyl groups (the common group R-CHO consisting of a carbonyl center bonded to hydrogen), able to form two Schiff bases.

MDA is a ubiquitously generated product since lipid rich bilayers - for example both plasma and mitochondrial membranes - are present in all cells and provide an optimal environment for producing a large abundance of this compound. Compared with reactive oxygen species (ROS), MDA has a relatively long half-life (minutes-hours) and a non-charged structure, which makes it a potentially more destructive compound. The reason is that MDA can affect cell structures, located in its vicinity, and also distant macromolecular targets from the MDA source [24].

Under physiological conditions, MDA is not a highly reactive compound, increasing its reactivity at lower pH. At strong acidic conditions, MDA can react with amino acids such as glycine, leucine, valine, and the guanidino group of arginine, to yield different adducts. In vitro experiments, with conditions more similar to physiological ones, showed that MDA can react with a broad variety of amino acids such as histidine, tyrosine, and arginine exclusively at the alpha amino group, and even cysteine. However, the reaction of MDA with cysteine at pH 7.4 is virtually non-existent. Findings from different experimental conditions indicate that MDA is low reactive, and its reaction with the alpha-amino groups of amino acids is in fact not favored [8,9,20]. In the reaction of MDA with secondary amines, the epsilon amino group of lysine is the main target [10,20]. In comparison with free amino acids, proteins seem to be more readily modified by MDA under physiological conditions, probably due to the more favorable (but undetermined) environmental conditions provided by proteins [9,10,20]. In vitro experiments incubating MDA with proteins, such as albumin and RNase, demonstrated that the free epsilon-amino group of lysine is the main target of MDA, although other amino acids like histidine, tyrosine, arginine, and methionine might also be altered to some extent [9,10,11,20]. In vivo, the main compounds detected, characterized, and quantified in proteins are the adduct MDALys and the cross-link lys-MDA-lys (see next sections). Figure 2 shows the lipid peroxidation process leading to the formation of MDA from the PUFA arachidonic acid, as well as reaction mechanisms of formation for the MDA-based adducts and cross-links with nucleophilic sites.

MDA can generate a diversity of adducts and intra- and inter-molecular cross-links resulting of the chemical and non-enzymatic modification of nucleophilic groups in macromolecules like proteins, but also nucleic acids and aminophospholipids (phosphatidylethanolamine (PE) and phosphatidylserine (PS)). Therefore, MDA can also react with the exocyclic amino groups of nucleosides to form alkylated products. Among DNA bases, the high nucleophilicity of guanine results in a higher vulnerability to generate adducts, MDA-deoxyguanosine (M1dG) being the most common [27,28,29]. Finally, MDA can also react with amino groups of aminophospholipids to generate adducts as MDA-PE [30]. Figure 2 shows a scheme for the formation of the MDALys adduct and the lys-MDA-lys cross-link by the reaction of malondialdehyde with lysine residues in proteins.

Considering the origin of MDALys, it is plausible to postulate that MDALys is an integrator biomarker of oxidative stress and lipid peroxidation [11,14,19,20,31].

## 3. Methods to Detect and Quantify MDALys in Proteins

Although the most used methods to detect MDALys adducts are based on mass spectrometry (MS)-based and antibody-based techniques (for more details, see [15]), other approaches which are currently in use are fluorescence-based methods [32,33], high-performance liquid chromatography (HPLC) [11], or nuclear magnetic resonance [13]. MS is the preferred method for the detection of MDA-dG detection, whereas MDA-aminophospholipids are analyzed using both fluorescence and MS based methods [30].

When an antibody-based technique is chosen, the recognition site is crucial. Thus, there are a number of antibodies (monoclonal and polyclonal) that recognized MDA-modified proteins which have been used for immunohistochemistry [34,35,36], immunoblot analysis [37,38,39,40,41], two-dimensional PAGE analysis [42], and ELISA [36,43,44]. All these approaches have the limitation that they do not per se provide information about the specific modified protein or the precise site of modification within the protein [15]. However, its excellent sensitivity allows a semi-quantitative approach to the degree of protein modification in a biological system, as well as visualizing the selectivity of modified proteins, which will require additional methods for their identification.

Among mass spectrometry (MS)-based techniques, gas chromatography-MS (GC/MS) has been extensively used to determine the steady-state level of MDALys at both subcellular and tissue levels (for references, see tables in next sections). Specifically, MDALys concentration was analyzed in a given sample (protein content 0.5–1 mg) as TFAME (trifluoroacetic acid methyl esters) derivatives. The samples were previously reduced with NaBH4 to stabilize the adducts to the conditions used. Then, acid hydrolysis was applied and derivatized samples were injected to GC coupled to an MS using an HP-5MS column (30 m × 0.25 mm × 0.25 μm), and a specific temperature program ranging from 110 °C to 300 °C. Finally, quantification was made by internal and external standardization using standard curves of deuterated and non-deuterated standards. If the analyses are performed using SIM-GC/MS (selected ion-monitoring GC/MS) the specific ions used for detection and quantification are lysine and *d*8-lysine, *m/z* 180, and 187, respectively; and MDAL and *d*8-MDALys, *m/z* 474 and 482, respectively. Thus, the amount of product can be expressed as mili- or micromoles of MDALys per mol of lysine. The high sensitivity, resolution, and throughput of this method allows an unambiguous detection of the MDALys adduct, as well as the precise quantification of the MDALys concentration in a biological system. However, this approach poses a limitation in that it does not provide information about the specific proteins that have been modified or the precise site of modification within the protein.

Other techniques, such as liquid chromatography-electrospray ionization or matrix-assisted laser desorption/ionization-MS (LC-ESI or MALDI-MS)-based proteomics analysis would be useful to overcome these limitations and obtain information about the specific site of MDA adducts formation [15,45]. This approach would be an excellent source of information in order to learn about the molecular mechanism, features related to specificity, and the meaning of the non-enzymatic modification. However, more advances in MS techniques are needed to shed light on this field, currently technically limited due to the low abundance of this non-enzymatic modification.

## 4. The MDA-Lipoxidized Proteome

Despite the fact that any protein is potentially a target for modification by MDA – as studies carried out in vitro in proteins, such as hemoglobin, albumin, RNase, insulin B-chain, spermidine, and ubiquitin, among others, seem to suggest – our knowledge on the MDA-modified proteome is currently very limited, and is restricted to studies basically carried out in human brain and plasma (Table 1). 

Although MDA-modified proteins are mainly located in the mitochondria, they are also described in other locations like nucleus, cytosol and cell membrane, as well as extracellular compartments such as plasma. These observations demonstrate the facility of migration of MDA due to their chemical traits.

The analysis of the MDA-lipoxidized proteome indicates that these modifications are not specific of a biological process or molecular function and suggest a wide-ranging effect of this product in cell structure and metabolism. Therefore, proteins involved in energy metabolism (glycolysis, TCA cycle, oxidative phosphorylation, energy transduction, and fatty acid beta-oxidation), cytoskeleton, neurotransmission, proteostasis, plasma transport, and structural components of extracellular matrix are modified (Table 1). These results reinforce the heterogeneity of this specific PTM and suggest that not all Lys residues can interact and react with MDA to generate MDALys. In line with this, it can be postulated that the specific structural traits and spatial location of Lys residues determine the generation of MDALys and, consequently, the molecular damage.

## 5. Metabolism of MDA-Modified Proteins 

The MDALys concentration of a particular cell type, or even of a specific subcellular compartment, tissue, organ or animal species, is the result of a complex system of interactions with the participation of multiple mechanisms. As a starting point, there are two determining factors: the lipid composition of a cell membrane, and the homeostasis of oxidative stress. In relation to membrane lipids, we know that there is a direct relationship between the degree of unsaturation and their susceptibility to lipid peroxidation [24]. Therefore, the cell tries to maintain the integrity of the membrane without giving up its composition by using defense, repair, and replacement systems to reduce its vulnerability and the impact of oxidative stress [55,56]. This will determine the MDA levels generated. Regarding oxidative stress, the net flow of free radicals generated at the mitochondrial level plays a key role since these free radicals are responsible for damaging cellular components and, among them, lipids. These two biological characteristics are determinants of the MDA levels generated [20,24], and both traits are related to the aging process and the determination of longevity [56,57,58,59].

Once MDA is formed, a cellular response to maintain basal concentration of this and other aldehydes within physiological limits is initiated. This adaptive response involves different mechanisms, such as enzyme-mediated detoxification, urinary excretion, and antioxidant responses [23,24]. In relation to the latter, carbonyl species can work by sending regulatory signals to activate specific protein targets in order to decrease lipoxidation-derived damage and improve antioxidant defenses. This adaptive response is, at least, partially mediated by the carbonyl compound hydroxynonenal that (i) modifies and activates the uncoupling proteins (UCPs), resulting in a reduction in mitochondrial ROS production [60]; and (ii) induces the activation of the antioxidant response signaling pathway Nrf2 that includes, among others, the expression of enzymes such as glutathione-S-transferase (GST), specifically designed to detoxify reactive carbonyl compounds, and GPx4 (phospholipid hydroperoxide glutathione peroxidase), designed to restore reduced states of membrane fatty acids from phospholipids to ensure membrane lipid homeostasis [61,62,63]. To date, this hydroxynonenal mediated antioxidant response is not described in MDA. In fact, the low or null reactivity of MDA with cysteine suggests that MDA does not have a regulatory signal activity. Although, the potential effects under special conditions of concentration or environment cannot be dismissed, it seems that MDA possesses a preponderantly cytotoxic role instead of a regulatory function. Furthermore, the non-reactivity of MDA toward glutathione significantly limits the capacity of enzymes designed to detoxify carbonyl compounds via conjugation with glutathione (GSH) to degrade it. Consequently, MDA demands alternative enzymatic detoxification ways. Among them are the enzymes aldehyde dehydrogenase and aldoketoreductases [20,64,65], which participate in the maintenance of MDA levels within physiological levels.

Despite these protective mechanisms, MDA reacts with lysine and generates MDALys in proteins and other cellular components. Specifically in protein terms, the degree of modification will be determined by numerous factors: Structural aspects of the protein related to the exposure and ease of access of MDA to the functional groups of amino acids capable of being modified, functional aspects derived from the microenvironment where the amino acid is located and is conditioned by the rest of amino acids in its immediate environment, the cellular location of the protein and proximity to potential sources of MDA, and its turnover rate. There is, however, a lack of studies analyzing MDALys site features. Regarding the latter, the turnover rate that is specific to the protein can also be conditioned by its regulation at the endocrine level. Therefore, it has been observed that glucocorticoids [66], insulin [67], and thyroid hormones [68] affect the steady-state level of MDALys by basically modifying the turnover rate at the cellular level, the membrane unsaturation and the mitochondrial free radical generation.

The fact that MDALys is an adduct or a cross-link depends on whether the protein can be degraded [69] or accumulated forming aggregates. In line with this, evidence showed that MDALys can be degraded, as it has been detected in the urine of various animal species, such as mice, rats and humans [20,69,70,71,72]. Therefore, the balance between all these factors will determine the concentration of MDALys in a given biological system.

## 6. Cytotoxic Effects of MDALys Adducts

The molecular consequences of MDALys adducts formation in proteins mostly include detrimental structural and functional changes. Thus, MDALys formation induces alterations in physico-chemical properties such as conformation [5], charge [73], and solubility [48], formation of intra- and inter-molecular protein cross-links and aggregates [11,48,51], loss of enzymatic activity [15,47], and accelerated rate (for MDALys) or resistance to proteolysis (cross-links) [69]. When biological effects are considered, deleterious consequences such as immunogenicity (MDA generates immunoreactive materials in proteins) [43,51], binder to the receptor for advanced glycation end-products (RAGE) [73], and induction of monocyte activation and vascular complications [74] have been described. Additionally, MDA-adducts formation on nucleic acids induces DNA damage and mutagenesis [27,28,29], as well as alterations in physico-chemical and biological properties of the lipid bilayer when aminophospholipids are modified [30]. As an in vivo example of their cytotoxic effects at cellular level, a relevant recent study demonstrated that MDA causes neuronal mitochondrial dysfunction by directly promoting ROS generation and modifying mitochondrial proteins [75].

## 7. Protein Lipoxidation by MDA in Physiological and Pathological Models

Previous data showed that MDA-lipoxidation is detected in all tissues analyzed using immunoblotting and MS techniques, although the degree of modification varies significantly among them. In line with this, the presence of MDALys was described at both mitochondrial and tissue levels in a diversity of animal species. A relation of tissues, includes the brain, heart, liver, kidney, skeletal muscle, and plasma. Table 2 shows the steady-state levels of MDALys measured by GC/MS in different tissues and species. Importantly, the steady-state level of MDALys is in the range of micromoles MDALys/mol lysine, clearly indicating the low abundance of this post-translational modification at the tissue level. Furthermore, the findings from the immunoblotting approach also confirm the selectivity of MDA-lipoxidized proteins.

Notably, the higher MDALys concentration seems to be more present at mitochondrial levels in comparison to the tissue as a whole, probably as an expression of specific traits in mitochondria (high lipid content, high degree of membrane unsaturation, and high free radical generation) favoring MDALys formation. Likewise, it is remarkable that the higher MDALys concentration is reported in long-lived tissues like brain and heart which, in addition, have higher energy demands, and share identical traits to mitochondria: High lipid content, high degree of unsaturation, and high flux of mitochondrial free radical generation. 

During the last 30 years, this post-translational modification has also been detected in several pathological models including metabolic diseases such as chronic iron overload [87], metabolic syndrome [88], and type 2 diabetes and its complications [44,52,89,90]; in vascular diseases like atherosclerosis [50,91,92]; and in a diversity of neurodegenerative diseases such as Alzheimer’s disease [34,42], Incidental Lewy Body Disease [35], Creutzfeldt-Jakob Disease [93], Pick’s disease [38], Lewy Body diseases [48], familial Parkinson’s disease [46], and X-adrenoleukodystrophy [94,95]. In all these cases the pathological state presented increased steady-state levels of MDALys ascribed to alteration in lipid profiles and/or oxidative stress.

Together, these findings indicate that increased MDALys content in tissue proteins is a direct consequence of increased intracellular MDA concentration. For this reason, MDALys can be proposed as a biomarker of lipoxidative stress.

## 8. Malondialdehyde-Lysine in Aging and Longevity

Studies based on different experimental paradigms link MDALys to aging and longevity. These paradigms include studies performed (i) On individuals during aging, (ii) using species with different longevity, or specific strains and mutants within a species with different species, and (iii) applying physiological treatments that modified the aging rate and longevity.

### 8.1. Changes in MDALys During Aging

There are several studies that directly relate the aging process with MDA lipoxidation-derived molecular damage accumulation (Table 3). Globally, this lipoxidation damage, measured in tissue homogenate and mitochondria, increases with age, with tissues composed of long-lived, postmitotic cells (brain, heart, and skeletal muscle) being the most affected. Notably, in reference to the brain, the accumulation of MDALys seems to be region-specific but the reasons for this fact remain to be elucidated. Reinforcing these findings, lipofuscin, a complex age-pigment derived from lipoxidation reactions (including reactions derived from MDA) and considered a hallmark of aging [32], also shows an accumulation that correlates with age [96,97]. This age-related accrual of molecular oxidative damage is concomitant with the increase PUFA content during aging described in several tissues and animal species, as well as detrimental changes in oxidative stress, both factors favoring lipid peroxidation and the subsequent formation of the lipoxidation derived products MDALys. This increase in MDALys concentration at the tissue level can also be extended to changes with age in MDA-dG [98], and MDA-aminophospholipids (reviewed in [30]). Importantly, the degree of change with age varies in a tissue-dependent way. The changes described could occur on the basis of the age-related changes in membrane physico-chemical properties, such as fluidity leading to an increased membrane rigidity and loss of function, which has been systematically described in diverse studies [30,99,100,101,102,103]. As a consequence of this accumulation of MDALys with age, and considering the selective pool of proteins that seem to be modified, it can be proposed that specific cellular biological processes such as energy production and proteostasis, biological functions, which require the participation of cytoskeleton, and functional properties of the extracellular matrix may be preferentially affected during aging by this nonenzymatic modification. The answer as to whether the link between MDALys and agingg is causation or correlation requires of additional studies, but it must be considered that MDALys is only one of many indicators of more widespread chemical damage in biological systems.

### 8.2. MDALys and Animal Longevity

The connection between lipoxidation-derived damage and animal longevity was first reported by Pamplona and collaborators [77]. The study demonstrated that the MDALys concentration of heart mitochondria from a long-lived pigeon was significantly lower than the one detected in a short-lived rat. Later, these results were observed in a wide range of tissues and animal species, including both vertebrates (mammals and birds) and invertebrates, where a lower concentration of MDALys at both, mitochondria and tissue was detected in long-lived vertebrates (birds and mammals), compared to short-lived ones (Table 4). In agreement with this, it was described that longevous species have a low degree of total tissue and mitochondrial fatty acid unsaturation resulting in a low sensitivity to in vivo and in vitro lipid peroxidation and a low mitochondrial free radical production [24,104,109]. Furthermore, lipofuscin also showed an accumulation rate that inversely correlates with longevity [97]. Altogether, these results suggest that the lipoxidative damage is an optimized feature associated with animal longevity.

In this context, the study of the steady-state levels of MDALys in physiological systems from exceptionally long-lived specimens is of particular relevance. Thus, in a recent work [108], the MDALys concentration of both brain and spleen from adult (28 weeks), old (76 weeks) and exceptionally old (128 weeks) BALB/c female mice was analyzed. The study described significantly lower levels of MDA protein damage in the brain and spleen from exceptionally old animals when compared to old specimens. Interestingly, the levels found in exceptionally old animals were in the same range of adult animals. Therefore, the maintenance of a MDALys concentration at an adult-like level could be key factors for achieving longevity. This is especially important if we consider that all animals have been fed the same diet so the differences observed in molecular damage and lipid composition were genetically regulated.

Flies (*D. melanogaster*) provide another example of variation in longevity within a species that extends to invertebrates previous findings in vertebrates. In this study, wild type strains of *D. melanogaster* of varying longevity and long-lived mutants were compared [110,111]. The results showed that the greater the longevity of the *Drosophila* strain, the lower the MDALys concentration. 

In additional comparative studies in invertebrate species using mutant worms of varying longevities [112], queen and worker honey bees [113], or bivalves with exceptional longevities [114], results certified that long-lived animals possess peroxidation-resistant membranes. However, in this case, there is a lack of information about the degree of MDA-lipoxidation-derived damage to their cellular components.

Interestingly, all these comparisons performed using animals with different longevity demonstrate a relationship between longevity, the membrane unsaturation and the content of lipid peroxidation products, being lower in long-lived animals compared to short-lived ones [59]. Yet, it is unknown if MDA-mediated damage, and particularly MDALys, is in agreement with the data offered by lipid composition and peroxidation.

Despite the limited information in potentially relevant comparative studies and animal models that differ in their longevities, all comparisons that have been currently carried out support the notion of an important role for membrane fatty acid composition and lipoxidation-derived molecular damage in the determination of longevity, and point out to MDALys as a potential biomarker of animal longevity.

### 8.3. MDALys in Experimental Studies of Longevity Extension by Nutritional and Pharmacological Interventions

The study of how “anti-aging” nutritional and pharmacological interventions affect the lipoxidation-derived molecular damage associated with aging rate and longevity is crucial to establishing a causative role for membranes, oxidative stress and lipoxidation-derived damage in the determination of longevity. In line with this, several studies applying dietary modification have been performed, and membrane fatty acid composition and lipoxidative-derived damage have been evaluated [116,117,118]. These studies were specifically designed to partially circumvent the homeostatic system of compensation of dietary-induced changes in membrane unsaturation which operates at cellular level. The findings obtained demonstrate that lowering the membrane unsaturation of cellular membranes protects tissues against lipid peroxidation and MDALys formation.

Available evidence from both nutritional and pharmacological interventions that extend longevity in experimental models reinforces the relationship between MDA-derived lipoxidation damage and longevity. So, dietary interventions such as caloric- (CR), protein- (PR) and methionine (MetR) restriction and pharmacological interventions such as rapamycin decreased the degree of membrane unsaturation (reviewed in [59,102,119,120]) and especially the level of MDALys in a variety of tissues (for instance, liver, heart, and brain) and animal species (mainly rodents) (Table 5). Furthermore, it has been demonstrated that CR is able to reduce the levels of lipofuscin in tissues of rodents and *C. elegans* [97,121,122,123,124].

These studies showed that although the changes observed in membrane unsaturation are moderate (2.5–10%), the reduction of lipoxidation-derived molecular damage is much larger (20–40%). This could be explained by other factors, such as the lower mitochondrial ROS generation, also induced by these nutritional interventions. Interestingly, these interventions showed a correlation between the magnitude of the dietary restriction applied and the changes observed in membrane unsaturation.

Therefore, the dietary manipulation and pharmacological intervention with anti-aging effects seem to trigger an adaptive response protecting the most basic requirements of membrane integrity and avoiding the cytotoxic effects of MDA-derived lipoxidation reaction, by attenuating the MDALys formation.

## 9. Conclusions

Non-enzymatic posttranslational modifications are an inexorable part of cellular metabolism. Malondialdehyde-lysine (MDALys) is an adduct with cytotoxic properties that integrates the degree of unsaturation of a biological membrane and oxidative stress. Among the wide variety of techniques and methods that allow its detection and quantification, immunological and mass spectrometry methods currently stand out. The low concentration of this modification in the different cellular components – found in a wide diversity of tissues and animal species – is indicative of the presence of a complex network of cellular protection measures to avoid its deleterious effects. The findings in different experimental paradigms such as: (i) Increase in the concentration of MDALys with age within an animal species, (ii) the presence of a lower concentration of MDALys the greater the longevity of the animal species, (iii) its accumulation above physiological levels in a wide variety of pathological conditions, and (iv) the downward modulation of its concentration by nutritional and pharmacological interventions that have shown an extension of longevity, make the MDALys adduct a potential biomarker of aging and longevity.

## Figures and Tables

**Figure 1 antioxidants-09-01132-f001:**
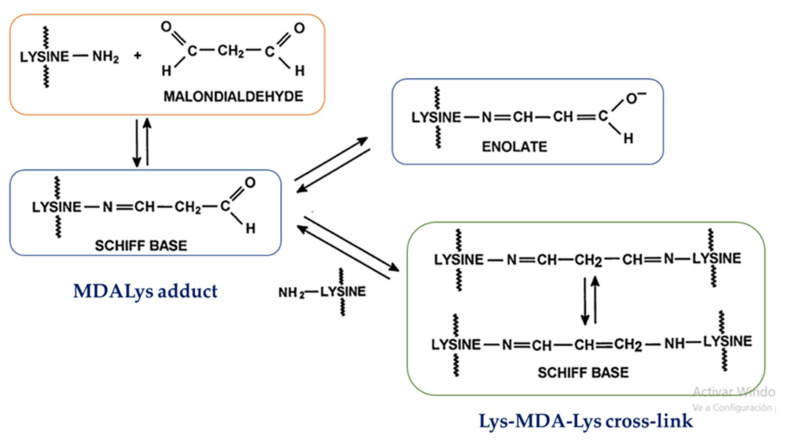
Formation of lysine-MDA adducts in protein. MDA is a dicarbonyl compound, which can form both mono- and di-Schiff-base adducts with lysine residues in protein, as well as enolate anions and other resonance structures. This is modified with permission from [11].

**Figure 2 antioxidants-09-01132-f002:**
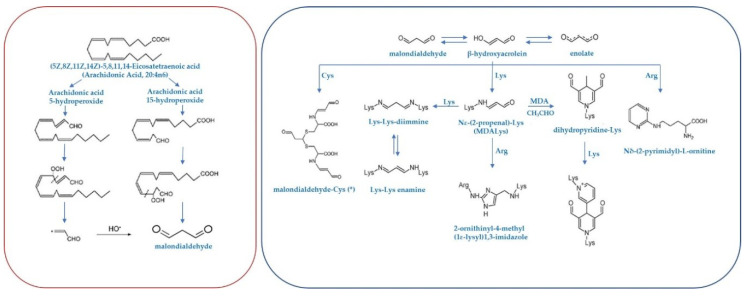
Mechanism of malondialdehyde formation from the polyunsaturated fatty acid arachidonic acid under oxidative conditions (modified with permission from [25]), and reaction mechanisms of formation of MDA-protein adducts and cross-links (modified with permission from [26]). Abbreviations: Arg, arginine; Cys, cysteine; Lys, lysine; MDA, malondialdehyde.

**Table 1 antioxidants-09-01132-t001:** Malondialdehyde-lipoxidized proteins identified by redox proteomics.

Protein	Species	Tissue	Main Location	Biological Process	Reference
alpha-Enolase	Human	Brain	Cytosol, cell membrane, nucleus	Energy metabolism (glycolysis)	[42]
gamma-Enolase	Human	Brain	Cytosol, cell membrane	Energy metabolism (glycolysis)	[42]
Gamma-Enolase	Human	Brain	Cytosol, cell membrane	Energy metabolism (glycolysis)	[46]
Aconitase	Mouse	Heart	Mitochondrion	Energy metabolism (TCA cycle)	[47]
Glutamate dehydrogenase 1	Human	Brain	Mitochondrion	Energy metabolism (TCA cycle)	[42]
alpha-ketoglutarate dehydrogenase	Mouse	Heart		Energy metabolism (TCA cycle)	[47]
Ubiquinol-cytochrome c reductase complex core protein 1	Human	Brain	Mitochondrion	Energy metabolism (ETC)	[42]
ATP synthase subunit beta	Human	Brain	Mitochondrion	Energy metabolism (OxPhos)	[42]
ATP synthase	Mouse	Heart	Mitochondrion	Energy metabolism (OxPhos)	[47]
Creatine kinase B-type	Human	Brain	Cytosol	Energy metabolism (energy transduction)	[42]
Very long chain acyl coenzyme A dehydrogenase	Mouse	Heart	Mitochondrion	Energy metabolism (Mitochondrial fatty acid beta-oxidation)	[47]
Dihydropyrimidinase-related protein 2	Human	Brain	Cytosol, cytoskeleton, membrane	Neurotransmission	[42]
Glutamine synthetase	Human	Brain	Cytosol, mitochondrion	Neurotransmission	[42]
Alpha-Synuclein	Human	Brain	Nucleus, cytoplasm, membrane, synapse, secreted	Neurotansmission	[48]
beta-Actin	Human	Brain	Cytosol (cytoskeleton)	Cytoskeleton	[42]
Glial fibrillary acidic protein	Human	Brain	Cytosol (cytoskeleton)	Cytoskeleton	[42]
Glial fibrillary acidic protein	Human	Brain	Cytosol (cytoskeleton)	Cytoskeleton	[46]
Glial fibrillary acidic protein	Human	Brain	Cytosol (cytoskeleton)	Cytoskeleton	[38]
Neurofilament light polypeptide	Human	Brain	Cytosol (cytoskeleton)	Cytoskeleton	[42]
Tubulin alpha 1B chain	Human	Brain	Cytosol (cytoskeleton)	Cytoskeleton	[42]
Tubulin beta chain	Human	Brain	Cytosol (cytoskeleton)	Cytoskeleton	[42]
Tubulin beta chain	Human	Brain	Cytosol (cytoskeleton)	Cytoskeleton	[46]
Vimentin	Human	Brain	Cytosol (cytoskeleton), nucleus	Cytoskeleton	[42]
Heat shock protein 60 KDa	Human	Brain	Mitochondrion	Proteostasis	[42]
Guanine nucleotide-binding protein G(I)/G(S)/G(T) subunit beta1	Human	Brain	Not described	Signal transduction	[42]
Low density lipoproteins (LDL)	Human	Plasma	Extracellular (plasma)	Lipid metabolism	[11,49,50,51,52]
Albumin	Human	Plasma	Extracellular (plasma)	Transport	[51]
Albumin	Human	Peritoneal dialysis fluid	Extracellular (plasma)	Transport	[53]
Collagen	Human	Cartilage	Extracellular matrix	Structural	[54]
Collagen	Human	Vascular system	Extracellular matrix	Structural	[8]
Fjbrinogen	Human	Plasma	Extracellular (plasma)	Coagulation	[51]

Main location, and biological process are based on what was reported in the UniProt database (http://www.uniprot.org/). Abbreviations: TCA cycle, tricarboxylic acid cycle; ETC, electron transport chain; OxPhos, oxidative phosphorylation.

**Table 2 antioxidants-09-01132-t002:** Steady-state levels of MDALys measured by GC/MS in different tissues and species.

Biological System	Animal Species	Concentration	Reference
Mitochondria (Brain)	Rat	571 ± 30	[76]
Mitochondria (Heart)	Pigeon	74 ± 5	[77]
Mitochondria (Heart)	Rat	452 ± 17	[78]
Mitochondria (Heart)	Rat	366 ± 18	[77]
Mitochondria (Heart)	Mouse	805 ± 60	[79]
Mitochondria (Kidney)	Rat	427 ± 17	[76]
Mitochondria (Liver)	Rat	387 ± 14	[78]
Mitochondria (S. Muscle)	Mouse	2357 ± 110	[80]
Mitochondria (S. Muscle)	Mouse	392 ± 29	[79]
Brain (whole)	Mouse	374 ± 23	[37]
Brain (whole)	Parakeet	305 ± 27	[37]
Brain (whole)	Canary	259 ± 22	[37]
Brain (whole)	Rat	337 ± 18	[81]
Brain (Amygdala)	Human	431 ± 32	[81]
Brain (Cerebellum)	Human	203 ± 20	[81]
Brain (Entorhinal cortex)	Human	283 ± 28	[81]
Brain (Frontal cortex)	Human	185 ± 12	[81]
Brain (Hippocampus)	Human	221 ± 25	[81]
Brain (Medulla oblongata)	Human	340 ± 19	[81]
Brain (Occipital cortex)	Human	219 ± 16	[81]
Brain (Spinal cord)	Human	352 ± 11	[81]
Brain (Striatum)	Human	450 ± 52	[81]
Brain (Substantia nigra)	Human	590 ± 29	[81]
Brain (Temporal cortex)	Human	164 ± 9	[81]
Brain (Thalamus)	Human	481 ± 42	[81]
Heart	Mouse	558 ± 22	[82]
Heart	Rat	381 ± 39	[82]
Heart	Guinea Pig	165 ± 36	[82]
Heart	Rabbit	143 ± 15	[82]
Heart	Sheep	129 ± 7	[82]
Heart	Pig	136 ± 22	[82]
Heart	Cow	113 ± 15	[82]
Heart	Horse	107 ± 7	[82]
Liver	Rat	184 ± 10	[67]
Liver	Mouse	274 ± 11	[83]
Kidney	Rat	228 ± 25	[84]
Plasma Low Density Lipoproteins (LDL)	Human	120	[11]
Skeletal muscle	Rat	282 ± 9	[85]
Skeletal muscle	Pigeon	239 ± 12	[85]
Whole fly	*D. melanogaster*	121 ± 2	[86]

Data are from healthy young/adult individuals or specimens. Units: µmol MDALys/mol lysine. Values are means ± SEM.

**Table 3 antioxidants-09-01132-t003:** Effect of aging on the steady-state level of MDALys in tissues from different species.

Experimental Model	Tissue	Change in MDALys Concentration with Aging	Method for MDALys Determination	References
Human (young adults vs. elderly subjects)	Hippocampus	Increased	IHQ	[34]
Rat (young vs. old animals)	Heart mitochondria	Increased	GC/MS	[104]
Rat (young, middle-age, old animals)	Liver mitochondria	Increased	GC/MS	[105]
Rat (adult vs. old animals)	Heart, liver	Increased	GC/MS	[106]
*D. melanogaster*	Whole fly	Increased	GC/MS	[86]
Rat (young vs. old)	Liver mitochondria	Increased	GC/MS	[107]
Mouse (young vs. old animals)	Brain, spleen	Increased	GC/MS	[108]
Mouse (young vs. old)	Heart mitochondria	Increased	GC/MS	[79]
Mouse (young vs. old)	Skeletal muscle mitochondria	Increased	GC/MS	[79]
Mouse (young vs. middle-age)	Liver	Increased	GC/MS	[83]
Human (adults vs. old subjects)	Frontal cortex Parietal cortex Cingulate gyrus Temporal cortex Entorhinal cortex Hippocampus Thalamus Caudate nucleus Putamen Visual cortex Substantia nigra Vermis	Increased Increased Unchanged Decreased Decreased Increased Increased Increased Increased Unchanged Unchanged Unchanged	WB	[41]
Rat (adult vs. old animals)	Kidney	Unchanged	GC/MS	[84]

Abbreviations: GC/MS, gas chromatography/mass spectrometry; IHQ, immunohistochemistry; WB, western blot.

**Table 4 antioxidants-09-01132-t004:** Relationship between lipoxidative damage measured as MDALys by GC/MS and animal longevity.

Animal Species	Longevity	Tissue (or Subcellular Organelle)	MDALys Concentration in Long-Lived Animal Species	References
Rat vs. pigeon	4 vs. 35 years	Heart mitochondria	Lower	[77]
7 mammalian species	From 3.5 years to 46 years	Liver	Lower	[115]
Rat vs. pigeon	4 vs. 35 years	Skeletal muscle	Lower	[85]
Mouse, parakeet, canary	3..5, 21, and 24 years	Brain	Lower	[37]
8 mammalian species	From 3.5 years to 46 years	Heart	Lower	[82]
*D. melanogaster* (long-lived mutant strains)	71 vs. 87 days	Whole fly and mitochondria	Lower	[110]
Exceptionally old mice	28, 76, and 128 weeks	Brain, spleen	Lower	[108]
*D. melanogaster* (short-lived Dahomey vs. long-lived Oregon R flies)	49 vs. 74 days	Whole fly	Lower	[111]

**Table 5 antioxidants-09-01132-t005:** Effect of pro-longevity and anti-aging nutritional and pharmacological interventions on MDALys concentration in different tissues and species.

Species	Tissue	DR Type (%)	DR Duration	Effect on MDALys	References
Rat	Liver mitochondria	CR 8.5%	7 weeks	Decreased	[125]
Rat	Liver mitochondria	CR 25%	7 weeks	Decreased	[125]
Rat	Heart mitochondria	CR 40%	4 months	Decreased	[126]
Rat	Heart mitochondria	CR 40%	1 year	Decreased	[127]
Rat	Liver mitochondria	CR 40%	22 months	Decreased	[105]
Rat	Liver	CR 40%	6 weeks	Decreased	[67]
Mouse	Liver	CR 40%	8 weeks	Decreased	[128]
Rat	Liver	PR 40%	7 weeks	Decreased	[129]
Rat	Liver mitochondria	MetR 40%	7 weeks	Decreased	[130]
Rat	Liver mitochondria	MetR 80%	7 weeks	Decreased	[78,130]
Rat	Heart mitochondria	MetR 80%	7 weeks	Decreased	[78]
Rat	Brain	MetR 80%	7 weeks	Decreased	[81]
Rat	Brain mitochondria	MetR 40%	7 weeks	Decreased	[131]
Rat	Kidney mitochondria	MetR 40%	7 weeks	Decreased	[131]
Rat	Heart mitochondria	MetR 40%	7 weeks	Decreased	[132]
Mouse	Brain	MetR 80%	4 months	Decreased	[120]
Rat	Liver mitochondria	MetR 40% at old age	7 weeks	Decreased	[107]
Rat	Heart mitochondria	MetR 40%	7 weeks	Decreased	[133]
Rat	Kidney	MetR 80% at old age	7 weeks	Unchanged	[84]
Pig	Liver mitochondria	MetR 30%	2 weeks	Unchanged	[134]
Rat	Liver mitochondria	Fasting	1 week	Increased	[135]
Mouse	Liver mitochondria	EOD (Every Other Day)	7 weeks	Decreased	[136]
Rat	Liver mitochondria	40% restriction of dietary amino acids (except methionine)	7 weeks	Decreased	[131]
Rat	Liver	Methionine dietary supplementation	7 weeks	Unchanged	[137]
Rat	Heart	Methionine dietary supplementation	7 weeks	Unchanged	[137]
Rat	Liver mitochondria	Cysteine dietary supplementation	7 weeks	Decreased	[138]
Rat	Liver	Corticosterone	4 weeks	Decreased	[66]
Rat	Liver	Thyroid Hormones	10 days	Decreased	[68]
Rat	Liver	Insulin	2 weeks	Increased	[67]
Rat	Liver	Growth hormone	2 weeks	Increased	[67]
Rat	Heart mitochondria	Atenolol	7 weeks	Decreased	[133]
Mouse	Heart mitochondria	Atenolol	16 months	Decreased	[79]
Mouse	Skeletal muscle mitochondria	Atenolol	16 months	Decreased	[79]
Mouse	Heart	Atenolol	2 weeks	Decreased	[139]
Mouse	Brain	Pioglitazone	2 months	Decreased	[95]
Mouse	Liver	Rapamycin	7 weeks	Decreased	[83]

Abbreviations: CR, caloric restriction; DR, dietary restriction; MetR, methionine restriction; PR, protein restriction.

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
