# Peer review of "The Advanced Lipoxidation End-Product Malondialdehyde-Lysine in Aging and Longevity"

_antioxidants, 2020, doi:10.3390/antiox9111132_

Round 1

Reviewer 1 Report

This is a review about MDA and MDA-lys.

  1. Unfortunately, there is a language problem and a lot of passages and sentences in the text are far from clear and hardly readable/understandable. Extensive editing of the English language throughout the text is required before any further action can be undertaken!! This manuscript should be completely rewritten and checked for language issues.
  2. The abstract is far from informative.
  3. It should be made a clear difference between MDA and MDA-lys. This is intermingled.
  4. What is the added value of MDA-lyine as biomarker in comparison to MDA?
  5. Is MDA-lys acid stable?
  6. An overview of all the reaction products of MDA with proytein, DNA and lipids would be very informative.
  7. I expect a critical appraisal of the literature about the detection of MDA-lys with different techniques and antibodies! This is just a summary of data.
  8. Which method of detection of MDA-lys is superior above others?
  9. Line 210: MDA as the only aldehyde with immunoreactivity is not true
  10. Line 212. Just one example of complete confusion: MDAlys formation on nucleic acids induces DNA damage and ….. Plenty of other examples in the text.…

Author Response

Responses to reviewers

R/ We would like to thank the reviewers for their interest, and for their helpful comments and suggestions. We have corrected the manuscript (MS) in order to address them, and we think that the MS is now improved. New text parts (or changes) throughout the MS are highlighted in yellow. The detailed answers to the specific points are given below.

Editor

 I would like to recommend you to include a chart where the different MDA-protein adducts are reported, not only Schiff bases.

R/ A new chart where the different MDA-protein adducts are reported has been now included in the new version of the MS as Figure 2. The former figure 2 is now figure 1, and the former figure 1 has now been removed.

Reviewer 1#

 This is a review about MDA and MDA-lys.

R/ We politely disagree with the reviewer’s comment. This work is a review specifically focused on MDA-lys. The information about MDA is limited to what we think is basic and necessary to understand the origin and meaning of MDA-lys.

Unfortunately, there is a language problem and a lot of passages and sentences in the text are far from clear and hardly readable/understandable. Extensive editing of the English language

throughout the text is required before any further action can be undertaken!! This manuscript should be completely rewritten and checked for language issues.

R/ We thank the reviewer for the suggestion. In accordance with the reviewer’s comment, we have extensively reviewed the language throughout the text with the help of a native expert. We hope that the new version of the MS will meet the requirements of the reviewer.

The abstract is far from informative.

R/ Some additions and modifications have been made to the abstract.

It should be made a clear difference between MDA and MDA-lys. This is intermingled.

R/ We politely disagree with the reviewer’s comment. We have been extremely scrupulous in making clear at all times when we refer to malondialdehyde, or when we are talking about malondialdehyde-lysine. We have not been able to detect throughout the work parts of the text that could lead to such confusion. .

What is the added value of MDA-lyine as biomarker in comparison to MDA?

R/ The objective of our review is not to determine whether one marker is better than the other. To our understanding, they are complementary markers that express different molecular realities, despite sharing common elements.

Is MDA-lys acid stable?

R/ In the procedure for the detection and quantification of MDA-lys using GC/MS from, e.g., a tissue homogenate, it is considered a step of reduction of the mda-lys adducts by borohydride in order to stabilize the link for the acid condition used for protein hydrolysis (HCl 6N) (Requena et al., 1997).

An overview of all the reaction products of MDA with protein, DNA and lipids would be very informative.

R/ The reader is provided with specific references if he / she wants to delve into the adducts that malondialdehyde generates at the level of DNA and lipids. We understand and reiterate that the work revolves around protein malondialdehyde modification and we do not want to disperse the reader’s attention. Hence, we ask the reviewer to allow us to maintain the current structure of the work.

I expect a critical appraisal of the literature about the detection of MDA-lys with different techniques and antibodies! This is just a summary of data.

R/ This section provides a brief overview of the current methods used to specifically detect MDALys and brief comments are made that point to the goals that can be achieved with them, as well as broadly mention their limitations. We understand that this work is not the place to discuss the pros and cons of different methodologies for determining MDALys.

Which method of detection of MDA-lys is superior above others?

R/ None is better than others. They are complementary and provide us with different information. This was covered in the methods section, which is not aimed at discussing the limitations and advantages of each of them.

 Line 210: MDA as the only aldehyde with immunoreactivity is not true

R/ We thank the reviewer for this appreciation. We have corrected the error in the new version of the MS.

Line 212. Just one example of complete confusion: MDAlys formation on

nucleic acids induces DNA damage and ….. Plenty of other examples in the

text.…

R/ In accordance with the reviewer’s comment we have corrected the indicated error. Furthermore, we have carefully reviewed the manuscript for possible additional errors and the error indicated by the reviewer is the only one we have been able to detect.

Reviewer 2#

We read with great interest this review describing the pathological role of malondialdehyde (MDA) and its methods of detection. This is a timely topic for neurodegenerative disorders such as Alzheimer's disease, traumatic brain injury and aging.

I think the authors have done excellent work; however I think there is a major part that is dedicated to the technical aspect of detection.

I recommend that the authors discuss the biological implication of this PTM and focus on  few diseases and the tables should be more informative as the authors are correlating malondialdehyde (MDA) to being biomarker; the tables should be separated  between human and  animal models.

What are the endpoints outcomes and levels of malondialdehyde (MDA) changes.

R/ We would like to thank the reviewer for the comments. However, we would like to express our bewilderment at them. The reasons are as follows:

  1. Our work is not focused on the pathological role of MDALys. The main goal of this review is to evaluate the relationship of MDALys with the physiological aging process, the determination of longevity, and the effect of anti-aging interventions.
  2. Great effort has been made to select only those works that detect and quantify mdalys in the physiological context of aging and longevity, avoiding at all times information related to pathological conditions.
  3. The technical aspects of detection only represent 7% of the main text of the review. So, this work is not methodological.
  4. We have dedicated a specific section to the molecular and biological effects of the formation of MDALys in proteins.

The review would benefit from a schematic Figure depicting the formation of MDA and how is it affecting brain proteins and how this can be translated in the nervous system to give this work a mechanistic arm. The major weakness is that the authors need to discuss the consequences of such PTM on the protein functions, expression and manifestation, lack of function, toxic function and how it is related to: AD, PD , brain injury and also aging.

R/ In accordance with the reviewer’s (and editor’s) comment, we have introduced a new figure (Figure 2) depicting the formation of MDA in the context of the lipid peroxidation process.

With respect to the role of the nonenzymatic PTMs in neurodegenerative diseases, we think that this consideration is outside our object, and in addition there are revisions where specifically these subjects are developed. Use as an example: Butterfield DA, Boyd-Kimball D. Mitochondrial Oxidative and Nitrosative Stress and Alzheimer Disease. Antioxidants (Basel). 2020 Sep 2;9(9):818. doi: 10.3390/antiox9090818; Pamplona R, Borras C, Jové M, Pradas I, Ferrer I, Viña J. Redox lipidomics to better understand brain aging and function. Free Radic Biol Med. 2019 Nov 20;144:310-321. doi: 10.1016/j.freeradbiomed.2019.03.016; Naudí A, Cabré R, Jové M, Ayala V, Gonzalo H, Portero-Otín M, Ferrer I, Pamplona R. Lipidomics of human brain aging and Alzheimer's disease pathology. Int Rev Neurobiol. 2015;122:133-89. doi: 10.1016/bs.irn.2015.05.008.

Reviewer 2 Report

We read with great interest this review describing the pathological role of malondialdehyde (MDA) and its methods of detection. This is a timely topic for neurodegenerative disorders such as Alzheimer's disease, traumatic brain injury and aging.

I think the authors have done excellent work; however I think there is a major part that is dedicated to the technical aspect of detection.

I recommend that the authors discuss the biological implication of this PTM and focus on  few diseases and the tables should be more informative as the authors are correlating malondialdehyde (MDA) to being biomarker; the tables should be separated  between human and  animal models.

what are the endpoints outcomes and levels of malondialdehyde (MDA) changes.

The review would benefit from a schematic Figure depicting the formation of MDA and how is it affecting brain proteins and how this can be translated in the nervous system to give this work a mechanistic arm.

The major weakness is that the authors need to discuss the consequences of such PTM on the protein functions, expression and manifestation, lack of function, toxic function and how it is related to: AD, PD , brain injury and also aging.

Author Response

(The authors gave the same response as above.)

Round 2

Reviewer 1 Report

Although  point of criticism  are well addressed, in my view a critical appraisal about the detection of MDA-lysine and the biological consequences of MDA-lysine  should be included to make it a more comprehensive review. As always, there are good and bad studies, performed with good and bad techniques.  The aim of a review is also to figure out what is meaningful and what is not because of bad performance of studies. That was also my major point of concern of the first version of the manuscript. Although the review is informative, unfortunately, there is still no critical appraisal of the literature in this review; it is just a long list of data.

Author Response

Reviewer 1#

Although point of criticism are well addressed, in my view a critical appraisal about the detection of MDA-lysine and the biological consequences of MDA-lysine should be included to make it a more comprehensive review. As always, there are good and bad studies, performed with good and bad techniques.  The aim of a review is also to figure out what is meaningful and what is not because of bad performance of studies. That was also my major point of concern of the first version of the manuscript. Although the review is informative, unfortunately, there is still no critical appraisal of the literature in this review; it is just a long list of data.

R/ In accordance with the comments from both editor and reviewer, we have introduced changes to clarify the aim of the review and modified the abstract to be more informative (see lines 11-23, and 41-44, also highlighted in yellow). We have also introduced new comments to stablish the limits of the techniques used (see lines: 117-119, 123-131, 141-149, also highlighted in yellow). Finally, comments about the biological consequences of mda-lys formation have also been included (see lines 294-301, 380-383). 

Reviewer 2 Report

Unfortunately, the authors have responded ina very arrogant tone and refused even to answer the basic requests by the two reviewers.

the work has a limited focus and they are clearly refusing to broaden the scope to discuss the biological aspects of the technical paper; the least it can be described.

I think this review lacks is so limited in focus and I don't think t is worth publishing in this journal.

thanks

Author Response

Reviewer 2#

Unfortunately, the authors have responded in a very arrogant tone and refused even to answer the basic requests by the two reviewers. The work has a limited focus and they are clearly refusing to broaden the scope to discuss the biological aspects of the technical paper; the least it can be described. I think this review lacks is so limited in focus and I don't think t is worth publishing in this journal.

Thanks

R/ We deeply regret that our responses led to misunderstanding. At no time has our intention been to show ourselves arrogant. Our answers were aimed to be correct and respectful at all times.

We understand that the reviewer argues that the review should open the focus of its theme, however, its aim focuses on the biological process of aging and longevity, and we understand that entering into pathological considerations completely distorts the meaning of the work. For this reason, we asked the reviewer to allow us to keep the focus of the work as it is posed. Indicate also that we have made changes to the work following the reviewer's recommendations. Thus, a new figure related to mda formation via lipid peroxidation has been introduced, and there is a section devoted to the specific cytotoxic effects induced by mda-lysine formation. We would like to express our apologies for our explanations leading to misinterpretations.

Round 3

Reviewer 1 Report

The authors have done an excellent job by addressing all minor and mojor points of concern and have revised the manuscript accordingly. For me, there are no additional point and i am satisfied with the manuscript as it is now. Well done.

Reviewer 2 Report

The work carries little value to ghe scientific feild with very limited focus